# Discovering Informative and Robust Positives for Video Domain Adaptation

**Chang Liu[1], Kunpeng Li[1], Michael Stopa[2], Jun Amano[2] & Yun Fu[1]**
[1]Northeastern University, Boston, MA, USA
[2]Konica Minolta, San Mateo, CA, USA
`liu.chang6@northeastern.edu, kinpeng.li.1994@gmail.com`
`{mstopa,jamano}@kmbs.konicaminolta.us, yunfu@ece.neu.edu`

## Abstract

Unsupervised domain adaptation for video recognition is challenging where the domain shift includes both spatial variations and temporal dynamics. Previous works have focused on exploring contrastive learning for cross-domain alignment. However, limited variations in intra-domain positives, false cross-domain positives, and false negatives hinder contrastive learning from fulfilling intra-domain discrimination and cross-domain closeness. This paper presents a non-contrastive learning framework without relying on negative samples for unsupervised video domain adaptation. To address the limited variations in intra-domain positives, we set unlabeled target videos as anchors and explored to mine "informative intra-domain positives" in the form of spatial/temporal augmentations and target nearest neighbors (NNs). To tackle the false cross-domain positives led by noisy pseudo-labels, we reversely set source videos as anchors and sample the synthesized target videos as "robust cross-domain positives" from an estimated target distribution, which are naturally more robust to the pseudo-label noise. Our approach is demonstrated to be superior to state-of-the-art methods through extensive experiments on several cross-domain action recognition benchmarks.

## 1 Introduction

Recent breakthroughs in deep neural networks have transformed numerous computer vision tasks, including tasks such as image and video recognition (He et al., 2016; Carreira & Zisserman, 2017a; Mittal et al., 2020). Nevertheless, achieving such remarkable results typically necessitates time-consuming human annotations. To address this issue, semi-supervised learning (Miyato et al., 2018) and self-supervised learning (SSL) (He et al., 2020) have been studied to utilize the knowledge available in a dataset with abundant labeled samples to improve the performance of models trained on datasets with scarce labeled data. However, the domain shift problem between the source and target datasets usually exists in real-world scenarios, leading to performance degradation. Unsupervised domain adaption (UDA) has been exploited to transfer knowledge across datasets with domain discrepancies to mitigate this problem. Although many methods have been created specifically for images, there is still a significant lack of exploration in the field of UDA for videos.

Recently, some studies have endeavored to perform UDA for video action recognition through the direct alignment of frame/clip-level features (Chen et al., 2019a; Pan et al., 2020a; Choi et al., 2020). However, these methods usually extend the image-based UDA methods without considering long-term temporal information or action semantics. Song et al. (2021) and Kim et al. (2021b) alleviate these issues with contrastive learning to learn such long-term spatial-temporal representations by instance discrimination. To further understand how contrastive learning helps UDA, we firstly recall that domain-wise discrimination and class-wise closeness are the two main criteria to solve UDA problems (Shi & Sha, 2012; Tang et al., 2020). Considering an unlabeled video from the target domain as an anchor, we explain that *Song et al. (2021) introduces intra-domain positives (in the form of cross-modal correspondence, e.g, optical flow) to help UDA by learning discriminative representation in the target domain.* Additionally, *Kim et al. (2021b) utilizes cross(source)-domain positives with the help of pseudo-labels and benefits the class-wise closeness by pushing the samples of the same class but different domains closer.*

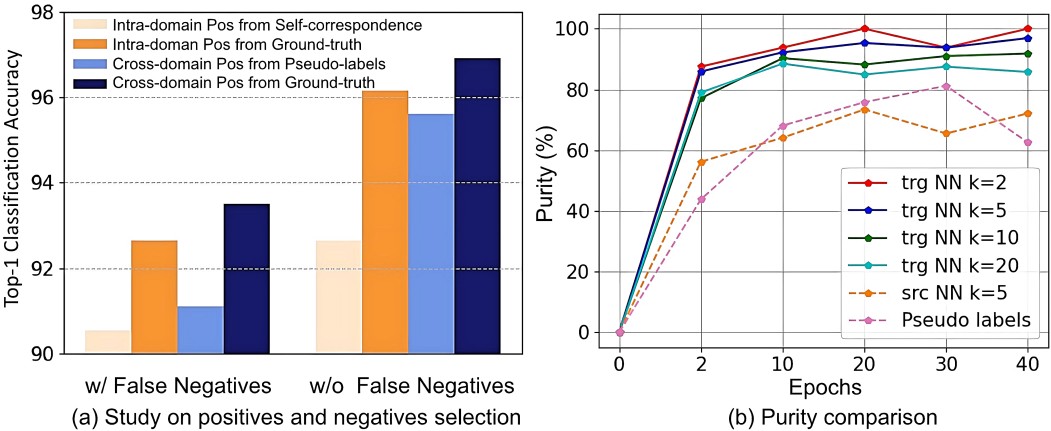

Figure 1: On HMDB→UCF: (a) **Effect of false negatives and intra/cross-domain positives.** We study two contrastive methods in video DA: STCDA (Song et al., 2021) with intra-domain positives and LCMC (Kim et al., 2021b) with cross-domain positives. We observe that *limited variations in intra-domain positives*, *pseudo-label noise in cross-domain positives* and *false negatives* are the three issues that largely hinder the performance. (b) **Purity comparison among nearest neighbors (NNs) and pseudo label.** We found the NNs in target domain are clean and fit as informative and robust intra-domain positives. Note that purity is the cleanness of pseudo-supervision compared to the ground truth.

By thoroughly studying the effect of the positives and negatives selection from the existing contrastive-based methods in Figure 1 (a), we empirically find that **limited variations** in intra-domain positives, **pseudo-label noise** in cross-domain positives and **false negatives** are the three issues that largely hinder the performance. Explicitly, when selecting the negatives based on the ground-truth, intra-domain positives from self-correspondence (Song et al., 2021) is 3.5% below the ones from ground truth. Further, selecting the cross-domain positives based on pseudo-labels induces 2% drop compared to the ground-truth. Importantly, when selecting the negatives from either different instance (Song et al., 2021) or pseudo-labels (Kim et al., 2021b), the performance of the four baselines in Figure 1 (a) drops by $2 \sim 4\%$. Besides, the purity analysis in Figure 1 (b) also indicates that the pseudo-labels of target video are noisy and thus not reliable for selecting cross-domain positives and negatives. Based on the observations, several straightforward questions might be raised:

1. Are there any unexplored intra-domain positives to enrich intra-domain variations?
2. How to alleviate the pseudo-label noise issue when selecting cross-domain positives?
3. How to address the adverse effect from false negatives?

To answer the first question, we propose to leverage the temporal and spatial augmentations of the unlabeled target video as intra-domain positives. The rationale is that partially modifying the spatial/temporal information of videos could alter the samples without changing the whole action semantics. Incorporating those samples as positives could help the model be invariant to spatial/temporal shift. Take a step further, we also explore the anchor's nearest neighbors (NNs) in target feature space, which capture rich target-domain variations. By analyzing the purity of the NNs in both source and target domain in Figure 1 (b), we observe that *target-domain NNs are clean and thus fit as intra-domain positives*.

To address the second question, we are motivated from (Xie et al., 2018) that *though the pseudo-labels assigned to target samples are noisy, the class-conditional centroids $\mu_c^t$ of target features would weaken this noise by averaging all the target features of same pseudo class*. To incorporate it into our contrastive learning framework, we re-formulate the optimization reversely by setting source videos as anchors and then mining target videos as cross-domain positives. We then estimate the gaussian target distribution $\mathcal{N}(\mu_c^t, \Sigma_c^t)$ of target features based on pseudo-labels. Given a source video as anchor, synthesized target features can be drawn from this distribution that shares the same class as the source anchor. Consequently, we could leverage those synthesized features as cross-domain positives as the estimated target distribution is robust to pseudo-label noise.

To tackle the last question, we present an effective non-contrastive learning framework without relying on negative samples for video domain adaptation. Specifically, we are motivated by the recent

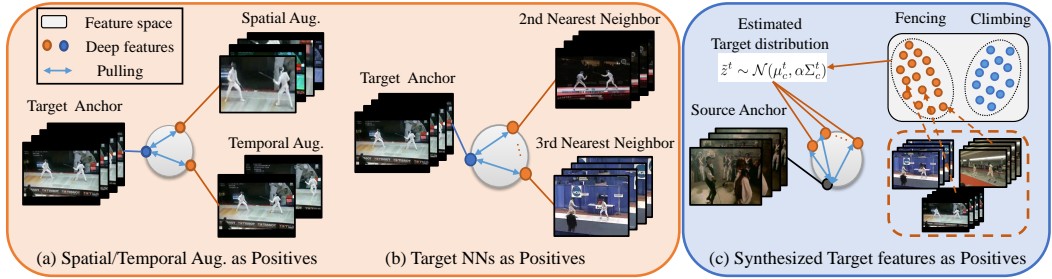

Figure 2: **Overview of our intra/cross-domain positives.** (a) spatial/temporal augmentations as intra-domain positives. (b) target nearest neighbors as robust intra-domain positives. Note that we consider self-augmentation as the 1st NN. (c) synthesizing target features from an estimated target distribution as cross-domain positives. In short, intra-domain positives help target-domain discrimination while cross-domain positives support class closeness across domains.

advances in SSL (Grill et al., 2020; Chen & He, 2021) to discard the negatives and maximize the similarity between the anchor and positives in feature space with the help of an extra MLP head and a stop-gradient operation. Beyond adopting this non-contrastive optimization into our setting, we highlight our compelling finding that *the MLP head is crucial for intra-domain positives optimization but largely hampers the convergence for cross-domain positives.* Consequently, we remove the MLP head for the latter case. To summarize our main contributions:

- By studying the effects of the positives and the negatives from previous contrastive-based video DA methods, we make the first attempt to emphasize that limited variations in intra-domain positives, pseudo-label noise in inter-domain positives and false negatives are the three under-explored bottlenecks

- We contribute a unified solution to address the above issues by introducing more informative and robust intra-domain and cross-domain positives without relying on negative samples for video DA problem.

- We conduct comprehensive experiments and analysis on challenging cross-domain action recognition benchmarks (UCF-HMDB and Epic-Kitchens) and thoroughly verify our superiority over state-of-the-art methods.

## 2 RELATED WORK

### 2.1 ACTION RECOGNITION

Action recognition is a challenging task with significant potential for practical applications. Many previous approaches are deployed in both 2D (Karpathy et al., 2014a), and 3D CNN-based framework (such as I3D (Carreira & Zisserman, 2017a), ResNet3D (Hara et al., 2017)), which achieves significant progress on RGB modality. To incorporate multiple modalities or temporal dynamics, the two-stream architecture (Simonyan & Zisserman, 2014) is a commonly employed technique in which RGB and optical-flow (Karpathy et al., 2014b) are processed independently using two CNNs, followed by a late fusion. The SlowFast (Feichtenhofer et al., 2019) utilizes dual branches to recognize actions through video with different frame rates. Although supervised benchmarks have shown encouraging results, such models heavily rely on large datasets that demand considerable human annotation effort. Conversely, our research is centered on UDA scenarios for action recognition, where the data in the target domain lacks labeling.

### 2.2 SELF-SUPERVISED LEARNING

Self-supervised representation has made significant strides in improving performance on images and videos through the use of various pretext tasks, such as rotations (Gidaris et al., 2018) and jigsaw puzzles (Noroozi & Favaro, 2016), as well as contrastive learning (Chen et al., 2020; He et al., 2020; Han et al., 2020). Recently, non-contrastive learning methods (i.e., BYOL (Grill et al., 2020) and SimSiam (Chen & He, 2021)) dominate the SSL performance by learning meaningful representation

by maximizing the similarity between two positive samples without considering negative pairs. While our method is motivated by the non-contrastive learning above, we highlight our difference in mining "informative" and "robust" intra-domain and cross-domain positives beyond instance-level variations for better target-domain discrimination and class-wise closeness. Importantly, we observe that the predictor in SimSiam (Chen & He, 2021), which is designed for preventing model collapse, would lead to convergence issue for cross-domain positives. In contrast, we remove the predictor for cross-domain positives in optimization while maintain it for intra-domain positives. We will explain the details in the experiment section.

### 2.3 Unsupervised Domain Adaptation

There are several directions to make UDA effective. Specifically, mainstream methods seek for learning domain-invariant feature representations by minimizing distribution discrepancy (Tzeng et al., 2014; Chung et al., 2022; Qiu et al., 2021; Liu et al., 2021), utilizing adversarial learning (Ganin et al., 2016; Long et al., 2018; Saito et al., 2018; Pan et al., 2020b), cross-domain augmentation (Wang et al., 2021; Li et al., 2021b) and contrastive alignment (Saito et al., 2020; Kim et al., 2021a; Harary et al., 2022; Li et al., 2021a). Self-training (Xie et al., 2018; Zou et al., 2018; 2019) also thrives on UDA by training the model with labeled source data and pseudo-labeled target data. However, less attention has been paid to video domain adaptation until recently. Existing video UDA methods can be summarized into two folds: temporal attention-based adversarial learning (Chen et al., 2019b; Luo et al., 2020; Pan et al., 2020a) and SSL-based cross-domain alignment(Munro & Damen, 2020; Choi et al., 2020; Song et al., 2021; Kim et al., 2021b). Explicitly, self-supervisions such as predicting video clips order (Choi et al., 2020), multi-modality correspondence (Munro & Damen, 2020) and contrastive learning (Song et al., 2021; Kim et al., 2021b; Sahoo et al., 2021), have been exploited. Compared to the image-based DA methods, our method provides a new perspective for domain discrimination and class-wise closeness via exploring positive samples in contrastive learning. Compared to the contrastive-based video DA methods, (1) we provide empirical insights on the limitations of existing contrastive methods by analyzing the effect of positives ans negatives; (2) we address the issues by proposing more informative and robust intra-domain positives and cross-domain positives without replaying on negative samples.

**Discussion on Xie et al. (2018).** It is worth noting some fundamental differences between our algorithm and Xie et al. (2018). From the **optimization view**, they propose centroids alignment across domains where target centroids are updated with the current target batch based on pseudo-labels and thus will receive gradients. If the pseudo labels are noisy, wrong gradients will hurt the training. In comparison, we align the cross-domain samples within constrastive framework where our synthesized target features are detached from gradients; thus, only the source batch will receive the robust gradients. **Algorithmically**, besides the estimation of target centroids, we further integrate target covariance to estimate the whole gaussian distribution of target domain. Table 2 shows that we have a significant improvement over Xie et al. (2018).

## 3 Proposed Method

The problem of UDA involves a labeled source domain $\mathbb{D}_s = (\boldsymbol{x}_i^s, \boldsymbol{y}_i^s)|i = 1^{N_s}$, comprising $N_s$ source videos, and an unlabeled target domain $\mathbb{D}_t = (\boldsymbol{x}_i^t)|i = 1^{N_t}$ of $N_t$ target videos. Both domains share a common label space $\mathcal{L}$. The joint distributions of source and target domain are not identically and independently distributed, specifically defined as $P(\boldsymbol{x}^s, \boldsymbol{y}^s) \neq Q(\boldsymbol{x}^t, \boldsymbol{y}^t)$. The objective is to develop a deep neural network for action recognition on both the labeled source videos $\mathbb{D}_s$ and unlabeled target videos $\mathbb{D}_t$. The model should be able to generalize effectively on new target domain videos. To specify, the model is comprised of a feature extractor $F$ followed by a classifier $G$.

### 3.1 Intra-domain Positives

The objective of intra-domain positives is to achieve domain-level discrimination, where the data in the source and target domains are segregated into separate semantic clusters.

**Intra-source Discrimination.** As source domain videos have labels, we simply minimize the supervised cross-entropy loss to achieve source-domain discrimination:

$$\mathcal{L}_{ce}(x_i^s, y_i^s) = -\frac{1}{N_s} \sum_{i=1}^{N_s} \log p_{i,y_i^s}, \tag{1}$$

where $p_{i,y_i^s} = G(F(x_i^s))_{y_i^s}$ is the $y_i^s$-th element of a K-dimensional prediction and K indicates the number of class.

**Self-supervision as Intra-domain positives.** Without proper regularization on unlabeled target data, target-domain discrimination cannot be realized for free. Motivated by the self-supervised learning methods (Chen & He, 2021; Bai et al., 2021), our first proposal is to train a model to learn a discriminative representation of the target domain by maximizing the similarity between different augmented views of the unlabeled target data. Specifically, we consider 1) spatial augmentation, including standard image-based augmentation such as random flip, crop and color distortion. 2) temporal augmentation, including video play speed (Benaim et al., 2020), and shuffled clips (Fernando et al., 2017). Precisely, we randomly double the sampling rate ($\times 2$) for video play speed or randomly shuffle a clip (1/3 consecutive frames) of a video.

The rationale behind is that *the effectiveness of the intra-domain positives are highly coupled with how "informative" and "clean" the positives are given an anchor video.* "Informative" positives usually capture rich spatial variations (e.g., background/viewpoint changes) or temporal dynamics (e.g., video speed/action length). Our choices of spatial and temporal augmentations above could partially change the anchor video along these variations without changing the whole action semantics. Consequently, incorporating those samples as intra-domain positives could help the model be invariant to the variations in spatial and temporal axis.

Explicitly, our model takes two randomly augmented views $x_1^t$ and $x_2^t$ from a target video $x^t$ as input: one view is strongly augmented by one of the augmentations above while the other view is weakly augmented. The feature extractor $F$ operates on both views, converting them into feature space. A two-layer prediction MLP head, represented by $H$, then takes the feature of one view, transforms it, and matches it with the feature of the other view. As shown in Figure 2 (a), our intra-target domain alignment loss can be formulated as maximizing the cosine similarity between the two views:

$$\mathcal{L}_{intra}(x_1^t, x_2^t) = \mathcal{D}(h_1^t, z_2^t), \tag{2}$$

$$\mathcal{D}(h_1^t, z_2^t) = -\frac{h_1^t}{\|h_1^t\|_2} \frac{z_2^t}{\|z_2^t\|_2}, \tag{3}$$

where $h_1^t = H(F(x_1^t))$ is the predictor output, $z_2^t = F(x_2^t)$ is the spatial-temporal feature, $\|\cdot\|$ is $l_2$-norm and $\mathcal{D}(\cdot, \cdot)$ is a cosine distance function. Specifically, the operation of stop-gradient is applied to $z_2^t$ and we implement it by modifying Eqn. 3 as $\mathcal{D}(h_1^t, stopgrad(z_2^t))$.

**Explanation on optimization without negatives.** Unlike traditional contrastive learning objective (Song et al., 2021; Kim et al., 2021b), we only involve positive pairs (e.g, $(x_1^t, x_2^t)$) into the optimization without negative samples. The insight (Chen & He, 2021) behind is that the negative samples in contrastive-learning play a role in preventing models from collapsing solutions (e.g, constant outputs). Alternatively, it is found that applying the predictor $H(\cdot)$ on one view $x_1^t$ and the stop-gradient operation $stopgrad(\cdot)$ on the other view $x_2^t$ could also avoid model collapse.

As the existing contrastive video DA methods are largely constrained by the false negatives issue (Figure 1), we leverage this optimization to cut off the reliance on the negatives and thus avoid the error accumulation led by false negatives.

**Target nearest neighbors as Intra-domain positives.** The empirical finding in Figure 1(b) shows that intra-domain NNs (target NNs) have higher purity than cross-domain NNs (source NNs) and pseudo-labels. It motivates us to further explore the intra-domain NNs as intra-domain positives which capture rich spatial and temporal variations in target domain.

Following the notation of Eqn. 3, we add the detached feature $z_2^t$ of the weakly-augmented view and its class pseudo label $\hat{y^t} = \arg\max G(z_2^t)$ to a memory bank, denoted as $\mathbb{M}_t = \{(z_i^t, \hat{y_i}^t)|_{i=1}^{N_t}\}$ in an online fashion where the size equals to the number of target dataset $N_t$. $\mathbb{M}_t$ is updated with the

newest features and pseudo labels given a current batch. Then we retrieve the $k$ nearest neighbors of $z_2^t$ in the memory bank to obtain $k$ target features $\{\hat{z}_j^t|_{j=1}^k\}$. Finally, we extend the Eqn. 3 to incorporate target NNs as intra-domain positives:

$$\mathcal{L}_{t-intra}(x^t) = \frac{1}{k} \sum_{j=1}^{k} \mathcal{D}(h_1^t, \hat{z}_j^t). \tag{4}$$

The overall loss is calculated by summing up the aforementioned loss function across all target videos $\mathbb{D}_t$. By pulling unlabeled target videos close to their NNs, the model could achieve the better target-domain discrimination. Importantly, the Eqn. 4 is equivalent to the Eqn. 3 when $k = 1$. As $k$ increases, the intra-domain positives could cover richer target-domain variations while under the risk of bringing false positives. The trade-off between the "informativeness" and "robustness" of intra-domain positives is discussed in experiment section in Figure 3.

### 3.2 ROBUST CROSS-DOMAIN POSITIVES

While the intra-domain positive focuses on domain-wise discrimination, it ignores the cross-domain alignment in the shared action semantic space. A natural solution is to extend Eqn. 4 by incorporating source domain positives which cover cross-domain spatial-temporal variations. As target videos do not have labels, we resort to use the pseudo-labels for choosing source samples or directly using source NNs as positives. However, as Figure 1 (b) suggests, both source NNs and pseudo labels have low purity. Incorporating cross-domain positives based on such noisy supervisions would result in false domain alignment.

To tackle this issue, we formulate it reversely to utilize source videos as anchors and mine for more informative target domain positives. Previous work (Xie et al., 2018) motivates us that, *though the pseudo-labels assigned to target samples are noisy, the class-conditional centroids $\mu_c^t$ of target features would weaken this noise by averaging all the target features of same pseudo class.* Consequently, the target centroid of each class $\mu_c^t$ is more robust to pseudo-label noise. To further explore target domain variations, we also estimate the corresponding covariance matrix $\Sigma_c^t$ from the target features of the same pseudo-class.

Instead of directly mining real target videos as positives based on noisy NNs or PL, we randomly sample a synthesized target feature from a normal distribution $\mathcal{N}(\mu_c^t, \Sigma_c^t)$ of the same action class $c$ to the source anchor video in the form of

$$\tilde{z}^t \sim \mathcal{N}(\mu_c^t, \alpha\Sigma_c^t), \tag{5}$$

where the coefficient $\alpha$ controls the target domain's variation. In the implementation, the target class centroids and the covariance matrix are computed based on the target features of the same pseudo-class from the memory bank $\mathbb{M}_t$ in Eqn. 4. As the estimation in the first few epochs is not so accurate, we set $\alpha = (t/T) \times \alpha_0$ where $t$ and $T$ are the current epoch and maximum epochs. As a result, the negative impact of the estimated covariance in the early training stage will be reduced.

Similarly to Eqn. 4, our model takes a source video $x^s$ as input (anchor) and processes it with the feature extractor $F$. Then we match it to the synthesized positives sampled from the class-conditional target distribution of class $y^s$ from Eqn. 5 in feature space. As shown in Figure 2 (c), when the number of sampled positives is $M$, our cross-domain alignment loss can be formulated as:

$$\mathcal{L}_{s-cross}(x^s) = \frac{1}{M} \sum_{i=1}^{M} \mathcal{D}(z^s, \tilde{z}_i^t), \tag{6}$$

where $z^s = F(x^s)$ is the feature of the source anchor and $\tilde{z}_i^t$ is the target positive sampling from $\mathcal{N}(\mu_{y_s}^t, \alpha\Sigma_{y_s}^t)$. We further derive a close-form upper-bound for the expectation of Eqn. 6 under all possible target features as $\mathcal{L}_{s-cross}^\infty$ in appendix B.1.

**Explanation on cross-domain optimization.** Unlike Eqn. 4, we do not transform the $z^s$ by the predictor $H$ which is originally designed for avoiding model collapse. Interestingly, we find that *discarding the predictor $H$ helps better convergence for cross-domain positives in optimization.* Our insight is that, as source feature $z^s$ and the synthesized target feature $\tilde{z}^t$ are from different distribution,

pulling them close requires non-trivial optimization efforts and thus models are less prone to collapse. In contrast, matching different views of a target video $z^t$ from the same distribution is relatively "easy". Without regularization from the predictor $H$, models will easily match all the views (e.g, augmentations or intra-domain NNs) of the target video into a constant output.

### 3.3 OVERALL OPTIMIZATION

In summary, the overall optimization includes the intra-domain positives in section 3.1 and the cross-domain positives in section 3.2 as follows:

$$\mathcal{L}_{all}^{rgb} = \mathcal{L}_{ce}^{s} + \mathcal{L}_{s-cross}^{\infty} + \lambda\mathcal{L}_{t-intra}, \tag{7}$$

where $\lambda$ is a hyperparameter to control the strength of intra-domain alignment. We empirically set the $\lambda$ to 15 for the convergence of $\mathcal{L}_{t-intra}$ due to the predictor. Our framework can be easily extended to two-stream network by adding Eqn. 8 for optical flow modality as $\mathcal{L}_{all}^{flow}$. Also, the optical flow correspondence of target videos can be integrated as intra-domain positives in $\mathcal{L}_{t-intra}$ (Eqn.15). Our algorithm with pseudo code will be found in appendix Alg.1.

## 4 EXPERIMENT

### 4.1 DATASETS

We conducted experiments to assess our method on two prevalent benchmark datasets for video domain adaptation, specifically UCF↔HMDB (Chen et al., 2019a) and Epic-Kitchens (Munro & Damen, 2020). To ensure consistency with prior research, we utilized the training and testing partitions provided by the respective authors in (Chen et al., 2019a; Munro & Damen, 2020).

**UCF↔ HMDB** is first assembled by Chen et al. (Chen et al., 2019a) for studying video domain adaptation problem. This dataset is a subset of the UCF (Soomro et al., 2012) and HMDB datasets (Kuehne et al., 2011), comprising $3,209$ videos distributed among 12 classes that overlap for human activity recognition. We present our results on two different scenarios: UCF→ HMDB and UCF← HMDB.

**Epic-Kitchens** is a fine-grained egocentric action recognition dataset with videos from three different domains (D1, D2, and D3) representing P08, P01, and P22 kitchens, respectively, from the full Epic-Kitchens dataset (Damen et al., 2018). The dataset is challenging and includes videos from the eight largest action classes. It is released by the authors in (Munro & Damen, 2020) for exploring video domain adaptation on the fined-grained setting.

### 4.2 COMPARISON WITH STATE-OF-THE-ART METHODS

**Baselines.** We conducted a comparative study with various baseline methods, including: (1) source only and supervised target only approaches, which utilize labeled source data and labeled target data, respectively, to train the neural network; (2) existing UDA methods that employ adversarial learning, such as DANN (Ganin et al., 2016) and ADDA (Tzeng et al., 2017), (3) existing UDA based on pseudo-labeling, including CRST (Zou et al., 2019) and MSTN (Xie et al., 2018). (4) existing video domain adaptation methods, including TA$^3$N (Chen et al., 2019a), ABG (Luo et al., 2020), TCoN (Pan et al., 2020a), MM-SADA (Munro & Damen, 2020), SAVA (Choi et al., 2020), STCDA (Song et al., 2021) and LCMC (Kim et al., 2021b).

**Results on UCF-HMDB.** Table 2 illustrates the performance of our method and other competing methods on the UCF-HMDB dataset. Our framework outperforms the previous state-of-the-art methods on various backbones and achieves the highest performance. Compared to the recent methods incorporating self-supervision such as clip-order prediction (SAVA (Choi et al., 2020)), appearance-motion correspondence (MM-SADA (Munro & Damen, 2020)) and contrastive learning (STCDA (Song et al., 2021) and LCMC (Kim et al., 2021b)), our method makes a significant improvement using I3D backbone with $\textbf{1.8\%}$ on UCF→HMDB and $\textbf{3.5\%}$ on HMDB→UCF task respectively. Though not specifically designed for motion space, our method can extend to two-stream backbone with additional gains to $\textbf{86.1\%}$ on UCF→HMDB and $\textbf{95.4\%}$ on HMDB→UCF, achieving the state-of-the-art performance.

Table 1: **Results on Epic-Kitchens Datasets.** All the methods use I3D two-stream backbone including both RGB and flow modality if not specify.

| Method | Backbone | D2 → D1 | D3 → D1 | D1 → D2 | D3 → D2 | D1 → D3 | D2 → D3 | Mean |
|---|---|---|---|---|---|---|---|---|
| Source-only | I3D | 42.5 | 44.3 | 42.0 | 56.3 | 41.2 | 46.5 | 45.5 |
| AdaBN (Li et al., 2018) | I3D | 44.6 | 47.8 | 47.0 | 54.7 | 40.3 | 48.8 | 47.2 |
| MMD (Long et al., 2015) | I3D | 43.1 | 48.3 | 46.6 | 55.2 | 39.2 | 48.5 | 46.8 |
| MCD (Saito et al., 2018) | I3D | 42.1 | 47.9 | 46.5 | 52.7 | 43.5 | 51.0 | 47.3 |
| MM-SADA (Munro & Damen, 2020)(RGB) | I3D | 41.7 | 42.1 | 45.0 | 48.4 | 39.7 | 46.1 | 43.9 |
| MM-SADA (Munro & Damen, 2020)(RGB + Flow) | I3D | 48.2 | 50.9 | 49.5 | 56.1 | 44.1 | 52.7 | 50.3 |
| LCMC (Kim et al., 2021b) | I3D | 49.5 | 51.5 | 50.3 | 56.3 | 46.3 | 52.0 | 51.0 |
| STCDA (Song et al., 2021)(RGB) | I3D | 44.4 | 41.1 | 47.7 | 45.5 | 41.2 | 47.6 | 44.6 |
| STCDA (Song et al., 2021)(RGB + Flow) | I3D | 49.0 | 52.6 | 52.0 | 55.6 | 45.5 | 52.5 | 51.2 |
| Ours (RGB) | I3D | 44.8 | 41.8 | 48.1 | 46.8 | 41.9 | 48.2 | 45.2 |
| Ours (RGB + Flow) | I3D | **49.9** | **53.5** | **52.7** | **57.5** | **46.9** | **53.4** | **52.3** |
| Supervised-target | I3D | 62.8 | 62.8 | 71.7 | 71.7 | 74.0 | 74.0 | 69.5 |

**Results on Epic-Kitchens.** Table 1 shows the results on Epic-Kitchens, which is another challenging dataset comprising a total of six transfer tasks with imbalanced data distribution across different action classes. Our approach demonstrates superior performance on all six transfer tasks with an average accuracy of **52.3%**. Compared to adversarial alignment (AdaBN (Li et al., 2018) and MCD (Saito et al., 2018) ), self-correspondence learning (MM-SADA (Munro & Damen, 2020)) and contrastive learning methods (STCDA (Song et al., 2021) and LCMC (Kim et al., 2021b)), similar observations can be found that our method outperforms all the competing methods in the challenging Epic-Kitchens dataset by **1.1%**.

## 4.3 ABLATION STUDY AND ANALYSIS

**Effectiveness of Proposed Positives.** As presented from Table 3, intra-domain positives bring a significant improvement on UCF→HMDB by **2.6%** and HMDB→UCF

Table 2: **Results on UCF-HMDB dataset.** Two-stream denotes the methods using I3D two-stream network for both RGB and flow stream respectively. Supervised Target denotes the baseline training with labeled target data only. * stands for the self-implementation.

| Method | Backbone | UCF → HMDB | HMDB → UCF |
|---|---|---|---|
| Source-only (Chen et al., 2019a) | ResNet-101 | 71.6 | 73.9 |
| DANN (Ganin et al., 2016) | ResNet-101 | 75.3 | 76.4 |
| JAN (Long et al., 2017) | ResNet-101 | 74.7 | 79.3 |
| AdaBN (Li et al., 2018) | ResNet-101 | 75.5 | 77.4 |
| MCD (Saito et al., 2018) | ResNet-101 | 74.4 | 79.3 |
| TA³N (Chen et al., 2019a) | ResNet-101 | 78.3 | 81.8 |
| ABG (Luo et al., 2020) | ResNet-101 | 79.1 | 85.1 |
| TCoN (Pan et al., 2020a) | ResNet-101 | 87.2 | 89.1 |
| Supervised Target (Chen et al., 2019a) | ResNet-101 | 82.8 | 94.9 |
| Source-only (Choi et al., 2020) | I3D | 80.3 | 88.8 |
| DANN (Ganin et al., 2016) | I3D | 80.7 | 88.0 |
| ADDA (Tzeng et al., 2017) | I3D | 79.1 | 88.4 |
| MSTN* (Xie et al., 2018) | I3D | 80.2 | 89.9 |
| CRST* (Long et al., 2017) | I3D | 80.9 | 90.2 |
| TA³N (Chen et al., 2019a) | I3D | 81.4 | 90.5 |
| SAVA (Choi et al., 2020) | I3D | 82.2 | 91.2 |
| STCDA (Song et al., 2021) | I3D | 81.9 | 91.9 |
| Ours | I3D | **84.0** | **94.7** |
| Supervised Target (Choi et al., 2020) | I3D | 95.0 | 96.8 |
| Source-only (Song et al., 2021) | Two-stream | 82.8 | 90.7 |
| MM-SADA (Munro & Damen, 2020) | Two-stream | 84.2 | 91.1 |
| STCDA (Song et al., 2021) | Two-stream | 83.1 | 92.1 |
| LCMC (Kim et al., 2021b) | Two-stream | 84.7 | 92.8 |
| Ours | Two-stream | **86.1** | **95.4** |
| Supervised Target (Song et al., 2021) | Two-stream | 95.8 | 97.7 |

by **5.0%**. Cross-domain positives also improve the performance by a large margin. Surprisingly, each proposed component individually outperforms the previous state-of-the-art methods (e.g., SAVA (Choi et al., 2020) and STCDA (Song et al., 2021)) on the I3D backbone. Incorporating these two components into the same framework brings further gains on both UCF→HMDB by **3.7%** and HMDB→UCF by **5.9%**. These results suggest that intra/cross-domain positives could benefit each other. Precisely, the better target domain discrimination brought by *intra-domain positives* helps the better estimation of target distribution in Eqn.5. In return, *cross-domain positives* help cross-domain alignment and result in more accurate target NNs for *intra-domain positives* in Eqn. 4. Last, the performance could be further boosted with our optimization without negatives.

**Effect of Optimization w/o Negatives.** As shown in Table 3, optimization without negatives boosts the performance of both intra-domain positive, cross-domain positives and jointly training by a significant margin. Explicitly, when jointly training, it improves **1.5%** on UCF→HMDB and **1.6%** on HMDB→UCF compared to the baseline of conventional contrastive optimization with negatives.

**Effect of Augmentation Strategies.** When $k = 1$, our *Intra-Pos* could generalize to self-supervised learning where the target anchor videos are pushed to be close to their augmentations. Explicitly, we incorporated spatial augmentations such as random cropping and color distortion, as well as temporal augmentations like video playrate change and clips shuffling. Table 4 shows that all types

of augmentations benefit the performance, and integrating all of them gives the best performance. Interestingly, UCF→HMDB favors the temporal augmentations more than the spatial augmentations while an opposite observation is found in HMDB→UCF.

Table 3: **Ablation Study on UCF-HMDB for each component. Intra-Pos**: Intra-domain Positives in Eqn.4, **Cross-Pos**: Cross-domain Positives in Eqn.6, **w/o Negatives**: Optimization w/o negatives in Sec.3.1

| Intra-Pos | Cross-Pos | w/o Neg | UCF → HMDB | HMDB → UCF |
|---|---|---|---|---|
| ✗ | ✗ | ✗ | 80.3 | 88.8 |
| ✓ | ✗ | ✗ | 81.8 | 92.6 |
| ✓ | ✗ | ✓ | 82.9 | 93.8 |
| ✗ | ✓ | ✗ | 80.9 | 90.9 |
| ✗ | ✓ | ✓ | 82.6 | 92.3 |
| ✓ | ✓ | ✗ | 82.5 | 93.1 |
| ✓ | ✓ | ✓ | **84.0** | **94.7** |

Table 4: **Ablation Study on augmentation strategies and cross-domain positive.** The gray rows are the **Intra-Pos** with all augmentations and the **Cross-Pos** with covariance estimation and w/o predictor.

| Method | UCF → HMDB | HMDB → UCF |
|---|---|---|
| Intra-Pos (default, k=1) | 81.8 | 92.3 |
| w/ Spatial Aug. | 80.8 | 92.1 |
| w/ Speed Aug. | 81.1 | 91.2 |
| w/ Shuffle Aug. | 81.3 | 91.4 |
| Cross-Pos | 82.6 | 92.3 |
| w/o $\Sigma_c^t$ | 82.2 | 91.8 |
| w/ predictor | 80.2 | 89.1 |

**Effect of Class-conditional Covariance.** Our *cross-domain* alignment pulls a source video to the synthesized target video representations from the estimated target distribution. We claim that the estimated class mean feature vector provides the shared action semantics from the target domain. Its corresponding covariance brings target spatial-temporal variations to the source samples. Table 4 shows a performance

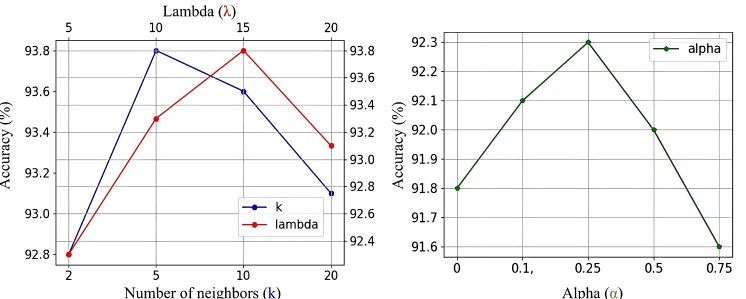

Figure 3: Hyperparameters sensitivity on the number of neighbors $k$ (bottom x-axis, left y-axis), intra-target coefficient $\lambda$ (top x-axis, right y-axis) and class covariance coefficient $\alpha_0$ on HMDB→UCF.

drop on *Cross-domain* without the covariance $\Sigma_c^t$ on both tasks of UCF-HMDB, demonstrating the importance of integrating target variations.

**Effect of Predictor in Cross-domain postives.** Unlike previous non-contrastive learning method (Chen & He, 2021), we find that applying the predictor results in poor performance in Tab. 4. We further tried increasing the optimization strength with larger loss ratio but still failed. *We hypothesize that, due to the domain shift, pulling the source features closer to the target requires non-trivial optimization effort and thus the model is less prone to the "collapsing solutions". Therefore, adding a predictor would make the optimization even harder.*

**Hyperparameters Sensitivity.** Figure 3 presents the effect of choosing different neighbors $k$ with $\lambda = 15$. We observe that adding neighbors in Eqn.4 generally boosts the performance and $k = 5$ gives the best performance. For $\lambda$, we set it $\lambda$ from $\{5, 10, 15, 20\}$ with $k = 5$. Empirically, we observe that relatively large $\lambda$ helps the convergence on the predictor in Figure 3, and $\lambda = 15$ achieves the best performance. For $\alpha_0$, it can be implicitly considered as the radius of target class distribution. It is shown that $\Sigma_c^t$ becomes most effective when $\alpha_0 = 0.25$ in Figure 3. In this case, the sampled target variations from $\Sigma_c^t$ are more likely to be clean. In summary, our method is robust to different hyperparameters.

## 5 CONCLUSION AND FUTURE WORK

In this work, we introduce the bottlenecks of in existing contrastive-based video DA methods and propose a unified solution to address them without relying on negatives by mining informative and robust intra-domain positives and cross-domain positives. Our approach has been thoroughly evaluated on various benchmark datasets, and the experimental results demonstrate its superiority over the current state-of-the-art methods. Regarding the future work, multimodal learning would be an interesting direction to be integrated with our algorithm by mining cross-modal nearest neighbors and estimating target distribution from other modalities.

ETHICS STATEMENT

In our paper, we strictly follow the ICLR ethical research standards and laws. All datasets we employed are publicly available, and all related publications and source codes are cited appropriately.

REPRODUCIBILITY STATEMENT

We highlight several components within this submission that effectively support the reproducibility of this work. **1)** Implementation details are provided in Appendix C, where the training dataset, backbone architecture, training schemes and hyperparameters are carefully stated. **2)** A more detailed pseudo code for our algorithm can be found in Alg. 1.

ACKNOWLEDGEMENT

This work was supported by Konica Minolta research funding.

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

# A APPENDIX

In this material, we show the additional optimization detail, comparison to Imagge-based Domain adaptation methods and algorithm pseudo code to our main submission.

# B OPTIMIZATION DETAIL

In summary, our framework can be readily implemented and optimized within an end-to-end deep learning framework using the overall objective function:

$$\mathcal{L}_{all}^{rgb} = \mathcal{L}_{s-ce}^{rgb} + \mathcal{L}_{s-cross}^{rgb}{}^{\infty} + \lambda \mathcal{L}_{t-intra}^{rgb}, \tag{8}$$

## B.1 CLOSE-FORM UPPER-BOUND FOR CROSS-POSITIVES OPTIMIZATION

The naive implementation is not computational efficient when $M$ is large. To tackle this issue, we consider the case that the number of $M$ goes to infinity, and then derive a close-form upper-bound for the expectation of Eqn. 6 under all possible target features as:

$$\mathcal{L}_{s-cross}^{\infty} = \mathbb{E}_{\tilde{z_i}^t}[\log e^{\mathcal{D}(z^s, \tilde{z_i}^t)}], \tag{9}$$

$$\leq -\log \mathbb{E}_{\tilde{z_i}^t}[e^{z^s \tilde{z_i}^t}], \tag{10}$$

$$= -\log e^{z^s \mu_{y_s}^t + \frac{\alpha}{2}(z^s)^T \Sigma_{y_s}^t z^s}, \tag{11}$$

$$= \mathcal{D}(z^s, \mu_{y_s}^t) - \frac{\alpha}{2}(z^s)^T \Sigma_{y_s}^t z^s. \tag{12}$$

For simplicity, all the $z^s$, $\tilde{z_i}^t$, $\mu_{y_s}^t$ and $\Sigma_{y_s}^t$ above are $l_2$-normalized if not coupled with $\mathcal{D}(\cdot, \cdot)$. The Eqn.10 emerges from the Jensen inequality $\mathbb{E}\log(X) \leq log\mathbb{E}[X]$ on the concave function $\log(\cdot)$. Based on the moment generation function $\mathbb{E}[e^{aX}] = e^{a\mu + \frac{1}{2}a^T \Sigma a}$ where $X \sim \mathcal{N}(\mu, \Sigma)$, Eqn.11 can be obtained from Eqn.10 under Gaussian assumption. The overall cross-domain alignment loss above is averaged for all source videos $N_s$. Notably, the computational cost of Eqn. 12 is different from backpropagating losses over $M \times N_s$ source-target pairs. As the $\mathcal{N}(\mu_{y_s}^t, \alpha\Sigma_{y_s}^t)$ is estimated offline based on the memory bank, Eqn. 12 can be computed efficiently without explicitly sampling target features.

## B.2 EXTENSION TO TWO-STREAM NETWORK

For fair comparison to the methods that utilizing optical flow such as MM-SADA Munro & Damen (2020), STCDA Song et al. (2021) and LCMC Kim et al. (2021b), our method could be extended to two-stream network using both RGB and optical flow as inputs as well. Following the setting above, we have:

$$\mathcal{L}_{all}^{flow} = \mathcal{L}_{s-ce}^{flow} + \mathcal{L}_{s-cross}^{flow}{}^{\infty} + \lambda \mathcal{L}_{t-intra}^{flow}. \tag{13}$$

Besides $\mathcal{L}_{all}^{rgb}$ and $\mathcal{L}_{all}^{flow}$, we also incorporate the cross-modality self-supervision in target domain as:

$$\mathcal{L}_{t-intra}^{cross}(x_t^{rgb}, x_t^{flow}) = \mathcal{D}(z_t^{rgb}, z_t^{flow}) + \mathcal{D}(z_t^{flow}, z_t^{rgb}), \tag{14}$$

where $z_t^{rgb} = F^r(x_t^{rgb}), z_t^{flow} = F^f(x_t^{flow})$ are the features of RGB and flow images.

To sum up, the overall objective function for two-stream network is shown as follows:

$$\mathcal{L}_{all} = \mathcal{L}_{all}^{rgb} + \mathcal{L}_{all}^{flow} + \lambda \mathcal{L}_{t-intra}^{cross}, \tag{15}$$

# C IMPLEMENTATION DETAILS

Following Choi et al. (2020); Song et al. (2021); Kim et al. (2021b), we use I3D Carreira & Zisserman (2017b) as the backbone feature encoder network, initialized with Kinetics pre-trained weights.

---

**Algorithm 1** Robust Cross-domain Positives for video DA

---

**Input:** Labeled source domain $\mathbb{D}_s = \{(\boldsymbol{x}_i^s, \boldsymbol{y}_i^s)|_{i=1}^{N_s}\}$, unlabeled target domain $\mathbb{D}_t = \{(\boldsymbol{x}_i^t)|_{i=1}^{N_t}\}$;
   warm-start iteration $T_0$, maximum iteration $T$ and batch size $B$; hyper-parameters: $\lambda$ and $\alpha_0$.
**Output:** Parameters of model $\boldsymbol{\Theta}_F$ and predictor $\boldsymbol{\Theta}_H$.
 1: **for** $t = 1$ to $T_0$ **do**
 2:    Update $\boldsymbol{\Theta}$ with labeled source videos $\mathcal{L}_{ce}^s$.
 3: **end for**
 4: **for** $t = T_0$ to $T$ **do**
 5:    $\alpha_= (t/T) \times \alpha_0$
 6:    Sample $\{(\boldsymbol{x}_i^s, \boldsymbol{y}_i^s)\}_{i=1}^B$ and $\{(\boldsymbol{x}_i^t)\}_{j=1}^B$ from $\mathbb{D}_s$ and $\mathbb{D}_t$, respectively.
 7:    Obtain deep features $\{\mathbf{z}_i^s\}_{i=1}^B$, $\{\mathbf{z}_j^t\}_{j=1}^B$ and softmax outputs $\{\hat{\boldsymbol{p}}_i^s\}_{i=1}^B$, $\{\hat{\boldsymbol{p}}_j^t\}_{j=1}^B$ for source and
       target samples, respectively.
 8:    Generate target pseudo labels based on $\{\hat{\boldsymbol{p}}_j^t\}_{j=1}^B$.
 9:    Update memory bank $\mathbb{M}_t = \{(z_i^t, \hat{y}_i^t)|_{i=1}^{N_t}\}$ with target features and pseudo labels in current
       batch $t$.
10:    Randomly apply strong spatial or temporal augmentation on the target samples as the second
       view $\{(\hat{\boldsymbol{x}}_i^t)\}_{j=1}^B$.
11:    Obtain their deep features and transform them as $\{\mathbf{h}_j^t\}_{j=1}^B$ by the predictor $\boldsymbol{\Theta}_H$.
12:    Given the current target features $\{\mathbf{z}_j^t\}_{j=1}^B$, we mine the top-K nearest neighbors from $\mathbb{M}_t$ as
       $\{\hat{z}_j^t|_{j=1}^k\}$ and compute the intra-domain positives loss $\mathcal{L}_{t-intra}$.
13:    **for** each class $c$ **do**
14:       Estimate target features means $\boldsymbol{\mu}_t^c$ according to memory module $\mathbb{M}_t$.
15:       Estimate the target intra-class covariance matrix $\boldsymbol{\Sigma}_t^c$ according to memory module $\mathbb{M}_t$.
16:    **end for**
17:    Based on $\mathcal{N}(\mu_c^t, \alpha\Sigma_c^t)$, compute the cross-domain alignment loss $\mathcal{L}_{s-cross}^\infty$.
18:    Update $\boldsymbol{\Theta}_F$, $\boldsymbol{\Theta}_H$ by minimizing the loss $\mathcal{L}_{all}^{rgb}$ in Eqn. 8 with stochastic gradient descent
       (SGD).
19: **end for**

---

The dimension of the features extracted from the I3D encoder is 1024. We randomly sample 16 consecutive frames out of a video clip with a size of $224 \times 224$. For inference, we use 16 uniformly sampled frames per video to recognize the action. We follow the standard 'pre-train then adapt' procedure used in prior works Tzeng et al. (2017); Choi et al. (2020) to train the model with only source data for 3 epochs as a warm start before the proposed approach is employed. Then we train models in Eqn. 8 for 40 epochs in total. For the hyperparameters, we select $\lambda$ from $\{5, 10, 15, 20\}$, $\alpha_0$ from $\{0.1, 0.25, 0.5, 0.75\}$ and the number of neighbors $k$ from $\{1, 2, 5, 10, 20\}$. We conduct hyperparameter sensitivity experiments in sec. 4.3. We set $k = 5$, $\lambda = 15$ and $\alpha_0 = 0.25$ for all datasets. We train all the models end-to-end using SGD with a momentum of 0.9 and a weight decay of 1e-7. We use an initial learning rate of 0.01 for the I3D with a cosine learning rate scheduler for our experiments. We use a batch size of 32 equally split over the two domains. We report the average action recognition accuracy over 3 random trials. We use four 12G NVIDIA GPUs for training.

## D    COMPARISON TO IMAGE-BASED DOMAIN ADAPTATION BASELINE

We compare our approach with the following Image-based DA baselines. (1) existing UDA based on advanced pseudo-labeling and entropy minimization methods which can be considered as cross(inter)-domain alignment approaches, including CRST Zou et al. (2019) and MSTN Xie et al. (2018) and MEDM Wu et al. (2021). (2) existing DA method based on intra-domain discrimination IDA Pan et al. (2020b).

We shows results of our method with other competing approaches on UCF-HMDB dataset in Table 5. Our framework outperforms both inter-domain and intra-domain methods by a significant margin.

Table 5: **Results on UCF-HMDB dataset for Image-based DA baselines.** Supervised Target denotes the baseline training with labeled target data only. * stands for the self-implementation.

| Method | Backbone | UCF → HMDB | HMDB → UCF |
|---|---|---|---|
| Source-only  Choi et al. (2020) | I3D | 80.3 | 88.8 |
| MSTN* Ganin et al. (2016) | I3D | 80.2 | 89.9 |
| CRST* Long et al. (2017) | I3D | 80.9 | 90.2 |
| MEDM* Wu et al. (2021) | I3D | 80.6 | 89.8 |
| IDA* Pan et al. (2020b) | I3D | 81.6 | 91.4 |
| Ours | I3D | **84.0** | **94.7** |
| Supervised Target Choi et al. (2020) | I3D | 95.0 | 96.8 |

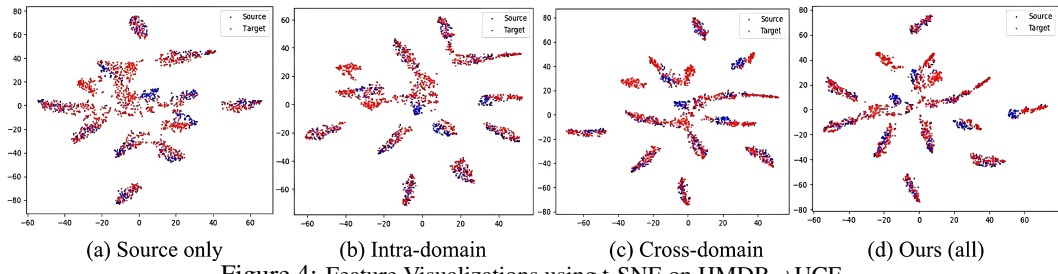

(a) Source only     (b) Intra-domain     (c) Cross-domain     (d) Ours (all)

Figure 4: Feature Visualizations using t-SNE on HMDB→UCF.

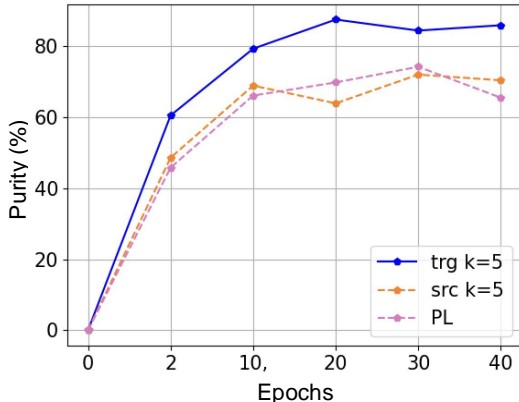

Figure 5: On UCF→HMDB: **Purity comparison among nearest neighbors (NNs) and pseudo label.**

## E ADDITIONAL ANALYSIS AND ABLATION STUDY

**Purity comparison on UCF→HMDB.** We conduct purity comparison on UCF→HMDB to further verify the observation in Figure 1(b). Given UCF→HMDB is a more challenging task than HMDB→UCF, we found similar observation that the target NNs are generally more robust and accurate than the source NNs and pseudo-labels.

Importantly, we first highlight our contribution in exploring the power of intra-domain positives which are missing in previous contrastive-based methods, such as Song et al. (2021) and Kim et al. (2021b). Further, we consider the target (intra-domain) NNs are generally more robust than PL and source NNs and suitable for intra-domain positives based on Figure 1(b) and Figure 5.

**Feature Visualizations.** We visualize the feature embedding by t-SNE (Maaten & Hinton, 2008) among each component of proposed methods, including the source only model, intra-domain, cross-domain and ours. As seen from Figure 4, intra-domain shows better class discrimination while cross-domain aligns two domains more tightly. Ours incorporates the merits of both two methods.

