# OpenReview forum: "Discovering Informative and Robust Positives for Video Domain Adaptation"
_ICLR.cc/2023/Conference — ICLR 2023 poster_

### Official Review · Reviewer_srUr · 2022-10-24

**Confidence:** 4
**Clarity, Quality, Novelty And Reproducibility:** It is easy to read and understand. Ho…
**Correctness:** 3
**Technical Novelty And Significance:** 2
**Empirical Novelty And Significance:** 2
**Recommendation:** 6

**Strength And Weaknesses:**


Strength
---
- The paper analyzes the noisy pseudo labeling for domain adaptation in action recognition and proposes a way to solve this with Gaussian data augmentations and target nearest neighbors.
- From a simple gaussian distribution, the method synthesizes the target features and uses these for cross-domain alignment.

Weakness
---
- The novelty is somewhat limited as the proposed components are borrowed from previous works (e.g., SimSiam). And the neighborhood clustering concept is also presented in [1], which is not referenced in this paper.
- Extensive comparison is needed. For example,  the benefit of synthesizing features is not very clear. How are the proposed cross-domain components compared to the previous cross-domain alignment methods? In addition, Instead of using the cross-domain alignment in this paper, how does the classical pseudo-labeling performs? How does the pseudo labeling in Sahoo et al. 2021 work compared to the cross-domain component in this paper? It is not clear that the proposed components are better than previous alignment methods.
-  As mentioned, the paper is missing some references for relevant works (e.g., feature generation [1], contrastive alignments for DA [2, 3, 4]).

[1] Li, Shuang, et al. "Transferable semantic augmentation for domain adaptation." *Proceedings of the IEEE/CVF Conference on Computer Vision and Pattern Recognition*
. 2021.
[2] Saito, Kuniaki, et al. "Universal domain adaptation through self supervision." *Advances in neural information processing systems*
 33 (2020): 16282-16292.
[3] Kim, Donghyun, et al. "Cds: Cross-domain self-supervised pre-training." *Proceedings of the IEEE/CVF International Conference on Computer Vision*
. 2021.
[4] Harary, Sivan, et al. "Unsupervised Domain Generalization by Learning a Bridge Across Domains." *Proceedings of the IEEE/CVF Conference on Computer Vision and Pattern Recognition*
. 2022.

**Summary Of The Paper:**

The paper handles noisy pseudo-labeling in domain adaptation for action recognition. The paper shows that the previous contrastive learning-based methods can be noisy and improves the model performance with robust cross-domain positives. Experiments are conducted on two benchmarks.


**Summary Of The Review:**

While the method outperforms existing SOTA methods, the novelty is somewhat limited since the proposed components are from previous works. And more extensive experiments are needed.

-- POST REBUTTAL --
Thank the authors for the response. My concerns are partially addressed. I still believe that the idea of neighborhood clustering is not new to domain adaptation. Even though there is a difference from NC[2], the proposed clustering does not seem very novel. Other implementation details seem to be engineering optimization with limited novelty. However, this work is meaningful as it provides extensive experimental results and analysis, which can contribute to the research community. Therefore, I increase my score to 6

---

> ### Author Response · Authors · 2022-11-17
> **Thanks you! Response to the Reviewer srUr, Part 1**
>
> Dear Reviewer srUr
>
> We sincerely appreciate the reviewer's recognition of our study and detailed comments to help us make further improvements for our submission. We summarize the reviewer’s comments and respond to them below.
>
> ### 1. The novelty is somewhat limited as the proposed components are borrowed from previous works (e.g., SimSiam).
> We acknowledge that we are motivated by SimSiam to solve the issue of the false negative, yet we would like to point out several key differences and findings:
> - In self-supervised learning (SSL), optimization with negatives (e.g., Moco, SimCLR) generally has similar performance on downstream tasks to the optimization without negatives (e.g, SimSiam, BYOL). We think that **SimCLR also suffers from false negatives issue, while SimSiam generally has a lower upper bound than SimCLR** as it is lacking negative information for a contrastive view. Consequently, two directions perform similarly in SSL tasks.
>
> - While in our Video DA task, we found that optimization without negatives has clear advantages over the other one, as shown in Table 3.  Specifically, it boosts Intra-Pos and Cross-Pos by 1~1.5% on average. These results indicate that **false negatives largely hamper the representation learning for cross-domain adaptation even though the SimSiam-like optimization has a lower upper bound, and this finding hasn't been discussed yet.**
>
> - More importantly, we observed that **directly optimizing the cross-positives without negatives (like SimSiam does) leads to the convergence issue**, as shown in **Table 4, w/  predictor.** Explicitly, it largely hampered the performance down by 2~2.8% on average. This number is almost similar to training with source data only. Instead, **we empirically found that removing the MLP head helps the convergence issue and makes the cross-positives contribute to the overall performance.**
>
> ### 2. The neighborhood clustering concept is also presented in [1], which is not referenced. Other relevant papers are also missing [1, 2, 3, 4].
>
> We kindly refer the reviewer to the **Related work, Section 2.3 UNSUPERVISED DOMAIN ADAPTATION, Line 4->cross-domain augmentation Li et al. (2021)** where we cited [1].
>
> Yet, if our understanding is correct, the reviewer might refer to the [2], which introduces the neighbor clustering concept. We appreciate it for pointing out these relevant papers [2,3,4] and have cited them in our new version (marked as blue).
>
> As we mentioned in the introduction, **domain-wise discriminativeness** and **class-wise closeness** are two goals to solve the DA problem. Compared to [2], our Intra-Pos and their Neighborhood Clustering (NC) [2] are both aimed at achieving Intra-domain discriminativeness in the DA problems. Yet we would like to clarify several key differences as follows:
>
> |  | NC [2] | Intra-Pos (ours) |
> | :--- | :------------: | ------------: |
> | __Data  flow__ | (1) a target sample as anchor; (2) all the target samples plus classifier weights as a memory bank  | (1) a target sample as anchor; (2) spatial/temporal augmentation; (3) k target samples (kNN)|
> | __Optimization__ | (1) Compute the similarity score between the anchor and all the samples in the memory bank; (2) Entropy minimization on this similarity score.| Non-contrastive alignment between the anchor and augmentation/target neighbors in feature space.|
> | __Insight__ | Pushing the anchor closer to all the potential Positives and away from all the potential negatives| Only pushing the anchor closer to Positives with high-purity (e.g, several kNN target samples and self-augmentation).|
>
> - We think NC[2] is more similar to contrastive learning-based methods such as Kim et al. (2021) and Song et al. (2021), where negative samples are involved.
> - Also, there is no sample selection process in [2] for positives and negatives which might contain pseudo-label noise, as we discussed in the paper. Instead, we select a few kNN and self-augmentations as positives which have higher supervision purity.

---

> > ### Author Response · Authors · 2022-11-17
> > **Response to the Reviewer srUr, Part 2**
> >
> > ### 3. Extensive comparison is needed. E.g, how are the proposed cross-domain components compared to the previous cross-domain alignment methods?
> > We kindly refer the reviewer to Table 1. We compared our method to previous cross-domain alignment methods and summarized them as follows:
> > - **Adversarial-based alignments** such as DANN Ganin et al. (2016), ADDA Tzeng et al. (2017), TA3N Chen et al. (2019a), TCoN Pan et al. (2020a) and MM-SADA Munro & Damen (2020).
> > - **Pseudo-labeling-based alignments**, such as CRST Zou et al. (2019), MSTN Xie et al. (2018), and LCMC Kim et al. (2021).
> > - **Self-supervised-based alignments**, such as Choi et al. (2020) and Munro & Damen (2020).
> > - **Contrastive-based alignments**, such as LCMC Kim et al. (2021) and STCDA Song et al. (2021).
> >
> > ### 4. Compared to the cross-domain component in this paper, how does the classical pseudo-labeling performs? How does the pseudo labeling in Sahoo et al. 2021 perform?
> > As discussed above, the classical pseudo-labeling methods such as CRST Zou et al. (2019), MSTN Xie et al. (2018) has been shown in Table 1.
> >
> > We would like to clarify that the pseudo-labelings (PL) in Sahoo et al. 2021 and LCMC Kim et al. (2021) are essentially similar. Specifically, both Eqn. (4) in Sahoo et al. 2021 and Eqn. (5) in Kim et al. (2021) followed the objective of supervised contrastive learning [A]. To modify it into the Video DA task, they both used pseudo-labels to assign positives and negatives.
> >
> > We compare our Cross-Pos to (1) the classical PL method (CRST Zou et al. (2019) and (2)the **pseudo-labeling component** of Sahoo et al. 2021, which we implemented and denoted as **Sahoo et al. 2021***. The results are shown as follows:
> >
> > |     Method     | UCF → HMDB |     HMDB → UCF              |
> > | ------------ | ----------- | -----------|
> > | __Zou et al. 2019__ |  80.9 |   90.2 |
> > | __Sahoo et al. 2021*__ |  81.3 |   90.6 |
> > | __Cross-Pos (ours)__     |  82.6 |  92.3          |
> >
> > Note:  For a fair comparison, our implementation of **Sahoo et al. 2021*** did not include temporal contrastive learning, background mixing, and an additional 3-layer graph convolution encoder as their original paper did.
> >
> > We want to emphasize that:
> > -  the major contribution **Sahoo et al. 2021** claimed is **Background Mixing** which they utilized an external preprocessing filter to extract the background of a source video and mix it with a target video.  By aligning the source and target at the pixel level, they achieve cross-domain alignment.
> > - They also added an extra **3-layer graph convolution encoder** on the backbone to further boost the performance.
> >
> >
> > [A] Prannay Khosla, Piotr Teterwak, Chen Wang, Aaron Sarna, Yonglong Tian, Phillip Isola, Aaron Maschinot, Ce Liu, and Dilip Krishnan. Supervised contrastive learning. In NeurIPS,2020. 5
> >
> > -----------------------------------
> >
> > **Lastly, thank you so much for helping us improve the paper and appreciate your open discussions! Please let us know if you have any further questions. We are actively available until the end of this rebuttal period. Looking forward to hearing back from you!**

---

> > > ### Author Response · Authors · 2022-11-30
> > > **Sincerely looking forward to your further feedback**
> > >
> > > Dear Reviewer srUr,
> > >
> > > Thanks for your valuable comments made in the review process, which are indispensable for us to polish the paper to a better version.  As you suggested, we have done the following:
> > >
> > > 1. Compare and clarify the difference between our work and previous works such as SimSiam and Neighbor clustering [2].
> > > 2. Modified the manuscript to add reference [1-4] into the updated version.
> > > 3. Compared our cross-positives (synthesizing features) to pseudo-labeling methods (Sahoo et al. 2021) empirically.
> > >
> > >
> > > [1] Li, Shuang, et al. "Transferable semantic augmentation for domain adaptation." Proceedings of the IEEE/CVF Conference on Computer Vision and Pattern Recognition. 2021.
> > > [2] Saito, Kuniaki, et al. "Universal domain adaptation through self supervision." Advances in neural information processing systems  33 (2020): 16282-16292.
> > > [3] Kim, Donghyun, et al. "Cds: Cross-domain self-supervised pre-training." Proceedings of the IEEE/CVF International Conference on Computer Vision . 2021.
> > > [4] Harary, Sivan, et al. "Unsupervised Domain Generalization by Learning a Bridge Across Domains." Proceedings of the IEEE/CVF Conference on Computer Vision and Pattern Recognition . 2022.
> > >
> > > **Please let us know if you have further comments so that we still have a chance to improve our paper before the deadline Dec 12th**. Given the discussion deadline is approaching, we really hope to have a further discussion with you to see if our responses solve your concerns. Thank you for being with us so far.
> > >
> > > Sincerely,
> > >
> > > Authors

---

> > > ### Author Response · Authors · 2022-12-08
> > > **Expecting feedback**
> > >
> > > Dear srUr,
> > >
> > > Sorry for keeping reminding you (since the discussion deadline of 12/12 is just around the corner). We are totally okay if your final decision is to keep the score. We are writing simply to remind you that we have made the clarification and experiments as you requested -- Do they make sense to you? And possibly, do they change your mind a bit?
> > >
> > > We value the feedback much more than a sole score. So we would really appreciate it if you could give us any feedback (like, if you think our point is not convincing, what kinds of experiments you think are missing to support the claim?). Your opinions are rather important for us to improve the work!
> > >
> > > Thank you!
> > >
> > > Sincerely,
> > >
> > > Authors

---

### Official Review · Reviewer_4bRo · 2022-10-24

**Confidence:** 4
**Correctness:** 3
**Technical Novelty And Significance:** 3
**Empirical Novelty And Significance:** 2
**Recommendation:** 6

**Clarity, Quality, Novelty And Reproducibility:**

Overall, the quality of this paper is good. The proposed methods are well-motivated in general, while part of the motivation requires a more detailed justification. The paper is well-written and easy to follow with certain novelty. Sufficient technical details are provided for future reproduction.

**Strength And Weaknesses:**

Strength:
This paper begins with an easy-to-follow analysis of what hinders existing contrastive-based methods, which motivates the following proposed methods. Overall, the proposed modifications are simple-yet-effective and well-motivated. The experimental results can further justify the effectiveness of the proposed methods.

Weakness:
Some details need to be further discussed.
a)	Authors claim that “intra-domain positives provide the least performance gains” indicates "their limitation in variations", while this causation is not convincing purely based on the comparison in Figure 1 (a). One could argue that such inferior performance is simply due to other reasons, e.g., the lack of minimizing domain discrepancy compared to cross-domain methods. It would be better for authors to provide more justification for this statement.
b)	While Figure 1 presents an intuitive analysis of the limitations of existing contrastive-based methods based on HMDB->UCF, I wonder whether these empirical observations also exist in other adaptation scenarios. More specifically, for Figure 1 (b), the higher purity of vanilla NN can be due to the fact that UCF is a relatively less challenging dataset (i.e., source only can already achieve 88.8). It is also known that multiple samples from UCF101 are clipped from the same raw video, which could also be the main reason why NN here is more accurate. Therefore, such observation on UCF101 may not work for other more challenging datasets.
c)	In Table 2, authors present the performance of their method utilizing RGB and RGB+Flow for comparison, while only introducing RGB+Flow results from previous methods. To make a fair comparison, authors should also, from my perspective, include the RGB results from previous methods in Table 2.
d)	Some minor formatting issues can be observed (e.g., Table 2 appears earlier than Table 1, and a citation error in Table 1 where MSTN^* Ganin et al. (2016) should be MSTN^* Xie et al. (2018)).

**Summary Of The Paper:**

This paper points out the existing limitations of current contrastive-based VDA methods, including limited variations of positives within the same domain, inaccurate cross-domain positive pairs, and inevitable false negatives. To address the above limitations, the authors introduce target-domain nearest neighbors and synthetic features from the target domain based on class centroids to perform VUDA in a non-contrastive manner. The proposed method is well-motivated, and its experimental results demonstrate its performance.

**Summary Of The Review:**

This paper presents a simple-yet-effective method for VDA, which leverage some machine-learning techniques to address the limitations of previous methods. The paper is presented clearly with sufficient details and analysis, while part of the motivation requires further justifications. Based on the quality of this paper, I would recommend a “marginally above the acceptance threshold”.

---

> ### Author Response · Authors · 2022-11-18
> **Thank you! Response to the Reviewer 4bRo**
>
> Dear Reviewer 4bRo
>
> We sincerely appreciate the reviewer's recognition of our study and detailed comments to help us make further improvements to our submission. We summarize the reviewer’s comments and respond to them below.
>
> ### 1. Justify the causality of the claim that “intra-domain positives provide the least performance gains” indicates "their limitation in variations"
> Thanks for pointing out the confusion on this statement. To justify the limited variations in intra-domain positives from Song et al. (2021), **we modify Figure 1(a) in the updated version** to show the performance of intra-domain positives with target domain variations based on ground truth (denoted as Intra-domain Pos from ground truth) and **revise the previous statement in Introduction marked as blue**.
>
> Compared to the self-correspondence (e.g, optical flow) as intra-domain positives from Song et al. (2021), adding variations in intra-domain positives could largely boost the performance by ~3.5%. We think this observation can better indicate the limited variations in intra-domain positives from the previous method. Hope this modification can address your concern.
>
> ### 2. Observation in Figure 1(b) on UCF101 may not work for other more challenging adaptation scenarios.
> To address your concern, we conduct an additional purity comparison on UCF→HMDB, which is a more challenging adaptation task than HMDB→UCF. Please refer to **Section E and Figure 5 in the updated supplementary**, where **we found similar observations that the target NNs are generally more robust and accurate than the source NNs and pseudo-label.**
>
> Additionally, we want to **emphasize our contribution in terms of exploring the power/limit of intra-domain positives
> which are missing in previous contrastive-based methods**, such as Song et al. (2021) and Kim et al.
> (2021b). Based on the findings in Figure 1(b) and Figure 5, we consider the target (intra-domain) NNs to be generally more robust than PL and thus suitable to serve as intra-domain positives.
>
> ### 3. Include the RGB results from previous methods in the Epic-Kitchens dataset.
> Please refer to Table 1 (Epic-Kitchens) in the updated version, where we added the RGB results from previous methods marked as blue.  For your information, Kim et al. (2021b) do not provide RGB results in their original paper. Thus we did not include the RGB results from their method.
>
> ### 4. Formatting issues
> Thanks a lot for helping us improve the paper. We have addressed these issues in the updated version.
>
> -----------------------------------
> **Lastly, thank you so much for helping us improve the paper, and appreciate your open discussions! Please let us know if you have any further questions. We are actively available until the end of this rebuttal period. Looking forward to hearing back from you!**

---

> > ### Comment · Reviewer_4bRo · 2022-11-19
> > **Respond to the response to my reviews**
> >
> > I would like to thank the authors for their effort in addressing my concerns and questions. I believe the authors have addressed them well and have clarified my doubts. I personally believe that this paper pushes the boundary of video domain adaptation, which is currently under-explored. Therefore I would vote positively for this paper to be accepted.

---

> > > ### Author Response · Authors · 2022-12-01
> > > **Thank you for your support and constructive comments**
> > >
> > > Dear Reviewer 4bRo,
> > >
> > > Thank you very much for your positive feedback and constructive suggestions, which indeed make our paper stronger!
> > >
> > > Plus, please feel free to discuss more if you have any other suggestions to improve the paper or potentially raise the score. We are actively available.
> > >
> > > Thanks again,
> > > Authors.

---

### Official Review · Reviewer_cfxm · 2022-10-25

**Confidence:** 4
**Correctness:** 3
**Technical Novelty And Significance:** 3
**Empirical Novelty And Significance:** 3
**Recommendation:** 6

**Clarity, Quality, Novelty And Reproducibility:**

The reviewer agrees that the work tackles an essential problem for video domain adaptation. However, several parts need further clarification by the authors to conclude the contributions of the work.

**Strength And Weaknesses:**

Strength:
- This paper has several interesting findings like "MLP head is crucial for intra-domain positives optimization but largely hampers the convergence for cross-domain positives". The proposed method is with adequate insights from these findings.

- This work introduces the bottlenecks of existing contrastive-based video DA methods.

Weakness:
- It seems that this work can be applied to existing video da models. I suggest the author should do further ablation study to testify whether this unified model can be applied to other video da models like TA3N.
- This work needs more ablation studies to support the proposed findings. Are these findings similar in different video DA benchmarks?
- This work only conducts experiments on two small cross-domain datasets UCF-HMDB and Kitchens. These two benchmarks only have a few classes. Validating the method in large-scale cross-domain datasets with more classes will make your conclusions more convincing.

**Summary Of The Paper:**

This work is the first to strengthen that limited variations in intra- domain positives, pseudo-label noise in inter-domain positives, and false negatives are the three under-explored key problems. Motivated by these concerns, this paper proposes a unified solution by introducing more informative and robust intra-domain and cross-domain positives without relying on negative samples for video DA. This model achieves state-of-the-art performance on several challenging cross-domain benchmarks for video DA.

**Summary Of The Review:**

This paper has proposed a novel contrastive-based unified video DA method without relying on negatives by mining informative and robust intra-domain positives and cross-domain positives.

---

> ### Author Response · Authors · 2022-11-19
> **Thank you! Response to the Reviewer cfxm**
>
> Dear Reviewer cfxm
>
> We sincerely appreciate the reviewer's recognition of our study and detailed comments to help us make further improvements to our submission. We summarize the reviewer’s comments and respond to them below.
>
> ### 1. Testify whether this model can be applied to existing video DA models like TA3N.
> We apply our model to TA3N, and the results are shown as follows:
>
> |     Method     | UCF → HMDB |     HMDB → UCF              |
> | ------------ | ----------- | -----------|
> | __TA3N__ |  81.4 |   90.5 |
> | __Ours__ |  84.0 |   94.7 |
> | __TA3N+Ours__     |  83.1 |  93.5          |
> |||
>
> **observation**:
> - Adding TA3N to our method would **decrease** the performance by ~1% on two tasks.
> - Our Cross-domain Positive alignment and TA3N have **similar functionality in terms of ensuring class-wise closeness across two domains**. It might explain why two methods cannot be complementary to each other.
>
> ### 2. More ablation studies to support the proposed findings in different video DA benchmarks.
>
> To recall our proposed findings in Figure 1,
> - In Figure 1(a), we first introduce that **limited variations in intra-domain positives**, **pseudo-label noise in cross-domain positives**, and **false negatives** are the **three bottlenecks** that hinder the performance of existing contrastive-based video DA methods .
> - In Figure 1(b), we found that NNs in the target domain are generally more robust than PL and thus suitable to serve as intra-domain positives
>
> To support our proposed findings in different video DA benchmarks,
> - For the 1st bullet above, We believe the proposed three bottlenecks generally exist for contrastive-based methods (**agnostic to datasets)** and could be fixed if the ground truth of intra-domain/cross-domain positives and negatives are given.
> - For the 2nd bullet, we **conducted an additional purity comparison on UCF→HMDB**, which is a more challenging adaptation benchmark than HMDB→UCF. Please refer to **Section E and Figure 5 in the updated supplementary** where **we found similar observations that the target NNs are generally more robust and accurate than the source NNs and pseudo-label**.
> - We notice that the reviewer points out our empirical finding that"MLP head is crucial for intra-domain positives optimization but largely hampers the convergence for cross-domain positives". We have shown this finding on UCF→HMDB and HMDB→UCF in Table 4. To further verify the finding, we **testify our Cross-domain Positives on the Epic-Kitchen dataset** as follows.
>
> | Cross-domain Positives   | Epic-Kitchen |
> | ----------- | ----------- |
> | w/o predictor | 50.7     |
> | w/   predictor     |  46.1     |
> |||
>
> - We have a similar observation that MLP/predictor hamper the convergence when optimizing cross-domain Positives.
>
>  `Please let us know if it addresses your concern on this point. We would love to conduct more experiments/analyses based on your feedback or request.`
>
> ### 3. Validate the method in other large-scale cross-domain datasets with more classes.
> For a fair comparison, we strictly follow the benchmarks and evaluation setting from the previous comparing methods, such as TA3N, Choi et al. (2020), Song et al. (2021), and Kim et al. (2021b) where UCF<->HMDB and Kitchens are evaluated.
>
> We agree with the reviewer that the community in video DA will definitely benefit from a new benchmark dataset with more classes. **The challenge is that, although there are several large-scale video recognition datasets (e.g, UCF101, HMDB) with many classes themselves, the **common/shared action classes** between datasets are usually limited**. We are actively looking forward to seeing a new benchmark in this community to come.
>
> Also, we would like to emphasize that the Epic-Kitchen dataset is an egocentric fined-grained action recognition dataset, which is very challenging (the performance of current methods is around 50%). Our method is effective in both UCF<->HMDB and Kitchens.
>
>
> -----------------------------------------
> **Lastly, thank you so much for helping us improve the paper, and appreciate your open discussions! Please let us know if you have any further questions. We are actively available until the end of this rebuttal period. Looking forward to hearing back from you!**

---

> > ### Author Response · Authors · 2022-11-30
> > **Sincerely looking forward to your further feedback**
> >
> > Dear Reviewer cfxm,
> >
> > Thanks for your valuable comments made in the review process, which are indispensable for us to polish the paper to a better version. As you suggested, we have done the following:
> >
> > 1. Combine our method with the existing video DA models such as TA3N and show the empirical results.
> > 2. More ablation studies to support the proposed findings in different video DA benchmarks.
> > - updated figure 1 to support findings of three bottlenecks;
> > - updated figure 5 to support why target NNs are suitable for intra-domain positives.
> > - support our empirical finding that"MLP head is crucial for intra-domain positives optimization but largely hampers the convergence for cross-domain positives" on the Epic-Kitchen dataset.
> > 3. Clarification on our choice of benchmarks compared to previous methods and comments on large-scale cross-domain datasets with more classes.
> >
> > **Please let us know if you have further comments so that we still have a chance to improve our paper before the deadline Dec 12th**. Given the discussion deadline is approaching, we really hope to have a further discussion with you to see if our responses solve your concerns. Thank you for being with us so far.
> >
> > Sincerely,
> >
> > Authors

---

### Official Review · Reviewer_9Kq9 · 2022-11-04

**Confidence:** 5
**Correctness:** 4
**Technical Novelty And Significance:** 3
**Empirical Novelty And Significance:** 3
**Recommendation:** 8

**Clarity, Quality, Novelty And Reproducibility:**

Overall the paper has explained the proposed idea with good clarity and performed extensive evaluation to show efficacy of the proposed approach.

**Strength And Weaknesses:**

**Strengths:**
- The paper raises some interesting points on the way state-of-the art models are trained for domain adaptive action recognition in a contrastive learning. The paper suggests that impurity of the pseudo-label and cross-domain FP/FN hinder contrastive learning, that results in a sub-optimal domain alignment.
- The proposed solution moves away from the contrastive learning and modifies it to include intra-domain positives and reduce reliance on negative samples. Furthermore, rather than following existing strategy of creating cross-domain positives, a new strategy is proposed that synthesizes positives for a source-domain anchor that likely reduces the pseudo-label noise.
- There are more minor observations for improvements such as value of MLP in contrastive learning for intra-domain and cross-domain optimization.
- Experimental analysis is exhaustive, includes state-of-the art methods for comparison and shows reasonable improvements with proposed strategy.
- Ablation studies also provide interesting insights into the individual component of the method and effects of hyper-parameters used on the final performance.


**Concerns:**
- One of the aspect of the method includes the use of fitting a Gaussian distribution on the target domain features, which is more or less similar to the strategy used here [1]. I would like to know more about how proposed method is different/better than [1] both intuitively and empirically. This would provide more information on the overall contribution of the work.
- The proposed solution revolves around reducing the FN/FP influence of the cross-domain alignment, but this intuition is never verified directly empirically. It would be beneficial to have FN/FP numbers before and after the use of proposed approach to further solidify the points raised in the introduction.
- Similarly, there should be an empirical analysis to show how the proposed synthesis strategy is reducing pseudo-label noise. The paper claims that such synthesis will weakens the noise, but is never empirically verified.

**Typos:**
- "robust intra-domain and cross-domain positives without *replying* on negative samples
for video DA problem." --> relying

**References:**
[1] Ding, Ning, et al. "Source-Free Domain Adaptation via Distribution Estimation." Proceedings of the IEEE/CVF Conference on Computer Vision and Pattern Recognition. 2022.




**Summary Of The Paper:**

The paper tackles unsupervised domain adaptation for the video recognition task. The paper raises concerns with the use of contrastive methods for aligning source and target domain features. Those concerns involve intra-domain positives, false positives in a cross-domain matches that hinder the contrastive learning process and negatively affect the adaptation. The proposed solution is to utilize intra-domain positives and mine robust cross-domain positives that reduces likelihood of pseudo-label noise. The proposed solution is shown to be superior than the state-of-the-art methods, by experiments on several cross-domain action recognition benchmarks.

**Summary Of The Review:**

Overall, the paper provides good contribution to the domain adaptation for the task of video action recognition. There are some concerns raised in the review, addressing them should provide more clarity on the final decision.

---

> ### Author Response · Authors · 2022-11-17
> **Thank you! Response to the Reviewer 9Kq9, Part 1**
>
> Dear Reviewer 9Kq9
>
> We sincerely appreciate the reviewer's recognition of our study and detailed comments to help us make further improvements to our submission. We summarize the reviewer’s comments and respond to them below.
> ### 1. How the proposed method is different/better than SDE [1], both intuitively and empirically.
>
> #### *Intuitively*
> | Method | SDE [1] | Ours |
> | :--- | :----: | ----: |
> | __Setting__ | Source-free DA where only source pretrained model and unlabeled target data are available | UDA where label source and unlabeled target data are available |
> | __Motivation__ | To estimate the source data distribution, which is missing in the Source-free DA setting | To reduce the adverse effect of the pseudo-label noise in unlabeled target data |
> | __Algorithm__ | Estimate the source data distribution $\mathcal{N}(\mu^s_c, \Sigma^s_c)$ from target data $\mathcal{N}(\mu^t_c, \Sigma^t_c)$  and utilize the previous UDA method (CDD[2]) to align the synthesized source and pseudo-labeled target data | Estimate the target data distribution $\mathcal{N}(\mu^t_c, \Sigma^t_c)$ from target data itself and align the labeled source data and the synthesized target data with non-contrastive optimization.|
> |  |  |  |
>
> - Though SDE[1] and our method both include Gaussian distribution estimation of target features $\mathcal{N}(\mu^t_c, \Sigma^t_c)$ , we are distinct in how we utilize this estimation, as the table above shows (**motivation/algorithm** row).
> - Essentially, SDE [1] does not propose any domain alignment method but estimates source data distribution which is missing in the Source-free DA setting, and thus any previous UDA alignment methods can be applied. Therefore, **SDE [1] is upper-bounded by the domain alignment approach they use (CDD [2] in their case)**.
> - In comparison, we propose an alignment approach to ensure intra-domain discriminative and cross-domain closeness. **Our method is orthogonal to SDE[1] as we have real source data available in our setting**.
>
>
> #### *Empirically*
> - **If comparing our method to SDE [1], we essentially compare it to CDD [2], which is the upper bound of SDE [1] with real source data available**.
>
> - Further, CDD [2] follows the same contrastive spirit of our comparing methods in Table 1, such as Song et al. (2021) and Kim et al. (2021). Specifically, CDD [2] assigns the pseudo labels to unlabeled target data and pushes the same class of source and target data closer and the different class of source and target data away. Similarly, CDD [2] also suffers from **no intra-domain positives**, **false cross-domain positives**, and **false negatives**.
>
> - For comparison, we implemented CDD[2] on our setting and recall the performance of Kim et al. (2021), which is a more recent contrastive-based alignment method than CDD[2] as follows:
> |Method|	UCF → HMDB|	HMDB → UCF|
> | :--- | :----: | ----: |
> |CDD[2]|	83.5 |91.2|
> |Kim et al. (2021)|	84.7 |92.8|
> |Ours| 86.1 |95.4|
> |  |  |  |
>
> [1] Ding, Ning, et al. "Source-Free Domain Adaptation via Distribution Estimation." Proceedings of the IEEE/CVF Conference on Computer Vision and Pattern Recognition. 2022.
>
> [2] Guoliang Kang, Lu Jiang, Yi Yang, and Alexander G Hauptmann. Contrastive adaptation network for unsupervised domain adaptation. In Proceedings of the IEEE Conference on Computer Vision and Pattern Recognition 2019.

---

> > ### Author Response · Authors · 2022-11-17
> > **Response to the Reviewer 9Kq9, Part 2**
> >
> > ### 2. Show the FN/FP numbers before and after the use of the proposed approach to further solidify the statements.
> > For recall, we claim that **the impurity of the pseudo-labels leads to false cross-domain positives and false negatives**, which hinder the existing contrastive-earning-based methods.  As a remedy, we tackle cross-domain FP by target distribution estimation and synthesize target features for alignment. Also, we conduct the optimization without relying on negatives to avoid using FN.
> >
> > To verify our intuition, we conduct an analysis with the source-only model on the HMDB → UCF task, where we set the number of positives to 1 and the number of negatives to 20.
> > - For all the training target samples, we show the percentage of cross-domain FP assigned by pseudo-labels (Before our method).
> > - We utilize our synthesis strategy to sample a synthesized target feature from $\mathcal{N}(\mu^t_c, \alpha \Sigma^t_c)$ for each source sample of the class $c$ where $\alpha=0.25$. We find the nearest neighbor (NN) of the synthesized target feature from all the target samples and assign the label of NN to it. Then, we verify whether this assigned label shares the same class (class $c$) as the corresponding source samples or not. We show the percentage of cross-domain FP after our method in the table below.
> > -  For all the training target samples, we show the percentage of FN if a target sample has been assigned at least 1 FN out of all 20 negatives (Before our method).
> > |Method|	Before our method|	After our method|
> > | :--- | :----: | ----: |
> > |Cross-domain False Positives |	21% | 5% |
> > |False Negatives| 28% | None|
> > | | | |
> >
> > We can see that:
> > - Before our method, for all the target samples, 21% of them had been assigned cross-domain FP, and 28% of them had been assigned FN.
> > - After our synthesis strategy, the cross-domain FP have been largely reduced (21%->5%). It is also worth noting that we could further reduce the FP by using smaller $\alpha$ in $\mathcal{N}(\mu^t_c, \alpha \Sigma^t_c)$ but at the sacrifice of cross-domain variations as shown in Table 4(b).
> > - For FN, as we utilize the optimization without negatives, we technically have 0% FN after our method. Yet, we also note that, even though we avoid using the negatives, our method would be inferior to the contrastive-based methods if they were given the **ground truth negatives**.
> >
> > ### 3. Verify how the proposed synthesis strategy is reducing pseudo-label noise.
> > As the pseudo-label noise leads to cross-domain false positives, our synthesis strategy can be verified to reduce pseudo-label noise if we show that our method could lead to a smaller number of cross-domain FP.
> >
> > We consider the analysis in Question 2 above could also verify this point. Specifically, our synthesized target features lead to 5% cross-domain FP compared to the 21% by noisy pseudo-labels.
> >
> > *Please let us know if it addresses your concern on this point. We would love to conduct more experiments/analyses based on your feedback or request.*
> >
> > -----------------------------------
> >
> > **Lastly, thank you so much for helping us improve the paper, and appreciate your open discussions! Please let us know if you have any further questions. We are actively available until the end of this rebuttal period. Looking forward to hearing back from you!**

---

> > > ### Author Response · Authors · 2022-11-30
> > > **Sincerely looking forward to your further feedback**
> > >
> > > Dear Reviewer 9Kq9,
> > >
> > > Thanks for your valuable comments made in the review process, which are indispensable for us to polish the paper to a better version. As you suggested, we have done the following:
> > >
> > > 1. Compare our method to [1] both intuitively and empirically.
> > > 2. Show the FN/FP numbers before and after the use of the proposed approach.
> > > 3. Verify how the proposed synthesis strategy is reducing pseudo-label noise
> > >
> > > **Please let us know if you have further comments so that we still have a chance to improve our paper before the deadline Dec 12th**. Given the discussion deadline is approaching, we really hope to have a further discussion with you to see if our responses solve your concerns. Thank you for being with us so far.
> > >
> > > Sincerely,
> > >
> > > Authors

---

> > > > ### Comment · Reviewer_9Kq9 · 2022-12-13
> > > > **Final Comments**
> > > >
> > > > Thanks to the authors for providing a detailed response. The arguments provided by the authors addresses all the points I have raised in my review. The paper provides an interesting solution which is well motivated and supported by extensive experiments. Hence, I am inclined to raise my ratings and recommend accepting the paper.

---

### Author Response · Authors · 2022-11-19
**To All Reviewers: Thank you and hope you may take a quick look at our responses**

Dear Reviewers,

*Thank you so much* for spending your precious time reviewing our paper. We have luckily got very informative feedback from your comments so far, which are indispensable for us to polish the paper to a better version.

Meanwhile, as you may notice, we have responded to all of you. We are now writing simply to wonder **if you could spend a few minutes taking a quick look at our responses in case we misread your comments or conducted the wrong experiments**. Also, based on our responses, **if you think there should be more experiments/analyses, please let us know *asap* so that we can have enough time to finish them early**.

We would also really appreciate it if you could give us a little more feedback based on our responses (e.g., *do our responses resolve your concerns?*)

Again, thank you all for being with us so far to improve the paper!

Sincerely,

Authors,11/18

---

### Author Response · Authors · 2022-12-08
**Dear ACs, Could you take a chance to remind the reviwers back to discussion? Thank you so much!**

Dear ACs and Reviewers,

Thanks a lot for arranging the reviewing process so far!

We genuinely thank all the reviewers for their very helpful comments. We have been pretty active in addressing their concerns and keeping adding new results in the past month. We would like to ask for more discussion, yet it seems the reviewers are a bit inactive in responding to our further feedback.

We deeply understand it is never an easy job to review a paper. But to ensure a fair evaluation of a paper, it might be better if the reviewers can further participate in the discussion. We are wondering if ACs could help with this. Thank you very much!

Sincerely,

Authors

---

### Author Response · Authors · 2022-12-08
**A bit more background that may help you evaluate the performance of this paper**

We summarize a bit the reviewing status after the discussions with reviewers, for the convenience of ACs and any readers who are interested in this paper. **We got 6/6/6/5. Reviewer 4bRo responded to us that our rebuttal had addressed his/her concerns well and would vote positively for this paper to be accepted while other reviewers haven't responded yet, unfortunately.**

**Reviewer srUr** is the most negative one. He/she rated 5 based on two parts: (1) limited novelty compared to SimSiam and Neighbor clustering (NC)[1]. (2) how are the proposed cross-domain components compared to the previous cross-domain alignment methods?

(1a) For comparison to SimSiam:
- We first observed that the SimSiam-like optimization (no negatives) shows advantages over the contrastive-based method (e.g, SimCLR) on the Cross-domain DA problem while they are generally comparable in the self-supervised learning task. This finding hasn't been explored yet.

- More importantly, we observed that **directly optimizing the cross-positives without negatives (like SimSiam does) leads to the convergence issue, as shown in Table 4, w/ predictor**. Explicitly, it largely hampered the performance down by 2~2.8% on average. This number is almost similar to training with source data only. Instead, **we empirically found that removing the MLP head helps the convergence issue and makes the cross-positives contribute to the overall performance**. Reviewer 9Kq9 and Reviewer cfxm also mentioned this contribution in their strengths.


(1b) For comparison to NC[1]:
- We highlight the difference in terms of data flow and optimization. In short, though naming neighbor clustering, the objective of NC[1] is more similar to contrastive learning, which involves the entropy minimization for **all the unlabeled samples (both positives and negatives)**.
- In comparison, our method emphasizes the importance of optimizing on **the informative and robust positives only** (a few positives).

(2) For comparison btw cross-domain components and previous domain alignment methods.
-  We have shown different kinds of domain alignment methods in Table 2 UCF-HMDB.
- By comparing our **cross-domain components only** in Table 2 and Table 4, our Cross-Pos outperform the top competitor of previous alignment methods by **0.6%**
- We also compare our **cross-domain components only** to Sahoo et al. 2021*(pseudo-labeling) and outperform it by ~1.5%.



**Reviewer 9Kq9** requested for comparison to SDE[2] and verifying how the proposed synthesis strategy is reducing pseudo-label noise (e.g, Showing the FN/FP numbers before and after the use of the proposed approach).
- We have replied as requested and would like to hear feedback on whether our response addresses your concern.
- Specifically, after our synthesis strategy, the cross-domain FP have been largely reduced (21%->5%)

**Reviewer cfxm** requested for an ablation study to verify (1) whether our model can be combined with the previous alignment method. (2) whether our findings hold on other video DA benchmarks. (3) More experiments on Video DA benchmark with more classes.
- We have replied as requested and would like to hear feedback on whether our response addresses your concern.
- Specifically, adding the previous alignment model TA3N to our method would decrease the performance by ~1% on two UCF-HMDB tasks.
- Our findings are still held on new video DA benchmarks such as HMDB->UCF and Epic-kitchen. Reviewer 4bRo was also concerned about this part and said our reply had addressed his/her concerns.
- For the challenge of Video DA benchmark with more classes: although there are several large-scale video recognition datasets (e.g, UCF101, HMDB) with many classes themselves, the common/shared action classes between datasets are usually limited. We are actively looking forward to seeing a new benchmark in this community to come. We know different researchers have their own standards. We just want the ACs and other readers to be aware of the benchmarking situation in the current video DA area and make fair evaluate.


Any comments about the above discussion are welcome!

Thanks,

Authors

 [1] Saito, Kuniaki, et al. "Universal domain adaptation through self supervision." Advances in neural information processing systems  33 (2020): 16282-16292
[2] Ding, Ning, et al. "Source-Free Domain Adaptation via Distribution Estimation." Proceedings of the IEEE/CVF Conference on Computer Vision and Pattern Recognition. 2022.

---

### Decision · Program_Chairs · 2023-01-20

**Decision:**

Accept: poster

**Justification For Why Not Higher Score:**

The whole approach is solid but more an engineering solution. The empirical benefit is sufficient for an ICLR poster, but the technical part is relatively weaker. In addition, the current presentation quality is not above the bar of ICLR and requires another revision.

**Justification For Why Not Lower Score:**

Reviewers are in unanimous agreement of accepting the paper, especially after rebuttal and discussion. While I can find some flaws that need the authors to further address, they are not serious enough to result in a rejection.

**Metareview: Summary, Strengths And Weaknesses:**

This paper proposed a solid engineering solution to video domain adaptation, a relatively under-explored problem compared to the image counterpart. The approach is built upon SimSiam but introduces several beneficial engineering improvements, based on exploring the informative and robust positives. The most important insights from this paper is that the video domain adaptation performance is hindered by "the limited variations in intra-domain positives, pseudo-label noise in cross-domain positives and false negatives". Through extensive evaluation with diverse baselines in video domain adaptation, domain alignment, self-training etc, as well as the extensive ablations of the proposed approach, the empirical benefits and claimed insights have been justified sufficiently.

The paper received mixed ratings. Reviewers pointed out essential issues which are with reasonable concerns in the AC's opinion. Authors made a great effort during the rebuttal, by providing detailed and relevant responses as well as a moderately revised version of the paper. After discussion (though after several rounds of expedition), two reviewers increased the overall score (5->6 and 6->8), while some other reviewer acknowledged the author response.

While the current version is acceptable, especially that the implementation details are included in the Appendix, AC personally has some further comments for the authors to address in a future version:

- The presentation is not nice enough, leaving with the impression of low-quality writing. For example, the citation format looks weird, in the form of Method Author (Year) throughout the paper. Authors shall read the guideline of ICLR and comply with the writing requirement. The correct form shall be Method (Author, Year) if the citation is not used as the subject of the sentence.

- Evaluation is only performed on UCF and HMDB, two obsolete datasets in the video domains. While I understand that in the DA community, previous work generally followed these two datasets, however, it is extremely easy to construct new datasets (even with identical source-target classes) from the big video datasets such as ActivityNet, Kinetics, and something-something. Since this is not done by the authors, we shall expect more sufficient evaluation on the UCF and HMDB. For example, as a video-oriented paper, we cannot see any showcases of video clips/snapshots regarding the insights of the algorithm, e.g. the discovered intra-domain positives, inter-domain positives, and etc.

- The quality of visualization in Fig. 3 is low-quality. We cannot observe any meaningful things from it. I suggest removing it directly. The vacated room can be used for a more in-depth case study. Fig. 1(b) is rasterized, why not use the vector format with higher quality. Fig. 1(a), text is overlapped with the bars...

- Discussion on Xie et al. (2018). This section shall be placed in the related work section or in the Appendix. Currently, it is placed in the body of the paper, making the presentation non-smooth.

- Necessary citations. For example, Eq. (9)--(12) seems to follow that in [Regularizing Deep Networks with Semantic Data Augmentation] or more older denoising auto-encoders.

Overall, authors have been very active in the rebuttal, by providing very detailed responses that overturn the reviewers' opinions. But as the AC, I consider the quality of the paper itself as the most crucial thing, and the current version is not satisfactory. Please make sure to improve and polish the paper significantly.

**Note From Pc:**

if the above contains the word "oral" or "spotlight" please see: "oral" presentation means -> notable-top-5% and "spotlight" means -> notable-top-25%. As stated in our emails, we are disassociating presentation type from AC recommendations